# The Exacerbating Effects of the Tumor Necrosis Factor in Cardiovascular Stenosis: Intimal Hyperplasia

**DOI:** 10.3390/cancers16071435

**Published:** 2024-04-08

**Authors:** Chandra Shekhar Boosani, Laxminarayana Burela

**Affiliations:** 1Somatic Cell and Genome Editing Center, Division of Animal Science, College of Agriculture Food and Natural Resources, University of Missouri, Columbia, MO 65211, USA; 2MU HealthCare, University of Missouri, Columbia, MO 65211, USA; 3Technology and Platform Development, Soma Life Science Solutions, Winston-Salem, NC 27103, USA; 4Aurora’s Degree and PG College, Chikkadpally, Hyderabad 500020, India; burelalaxminarayana@adc.edu.in

**Keywords:** epigenetics, TNF-α, inflammation, vascular restenosis, neointimal hyperplasia, vascular smooth muscle cells, proliferation, migration, phenotype switch

## Abstract

**Simple Summary:**

Targeting TNF-α was found to be helpful in alleviating inflammation. With TNF-α being a master regulator of inflammation, drugs that target TNF-α can help treat many diseases such as cancer and cardiovascular diseases, where TNF-α plays a critical role. Currently, anti-TNF-α drugs are not approved for treating vascular restenosis. This review article highlights the specific role of TNF-α in promoting proliferation, migration, phenotype switch, and cellular processes in vascular smooth muscle cells, which are the basis for restenosis. Molecular pathways and other mediators associated with TNF-α-induced mechanisms are discussed, which may help to develop a better strategy to use TNF-α antagonists for the treatment of restenosis.

**Abstract:**

TNF-α functions as a master regulator of inflammation, and it plays a prominent role in several immunological diseases. By promoting important cellular mechanisms, such as cell proliferation, migration, and phenotype switch, TNF-α induces its exacerbating effects, which are the underlying cause of many proliferative diseases such as cancer and cardiovascular disease. TNF-α primarily alters the immune component of the disease, which subsequently affects normal functioning of the cells. Monoclonal antibodies and synthetic drugs that can target TNF-α and impair its effects have been developed and are currently used in the treatment of a few select human diseases. Vascular restenosis is a proliferative disorder that is initiated by immunological mechanisms. In this review, the role of TNF-α in exacerbating restenosis resulting from neointimal hyperplasia, as well as molecular mechanisms and cellular processes affected or induced by TNF-α, are discussed. As TNF-α-targeting drugs are currently not approved for the treatment of restenosis, the summation of the topics discussed here is anticipated to provide information that can emphasize on the use of TNF-α-targeting drug candidates to prevent vascular restenosis.

## 1. Introduction

In several disease conditions, unregulated or uncontrolled inflammation is seen as a major significant factor that drives the disease state. However, in normal growth and health conditions, the inflammation induced is highly regulated, as seen during angiogenesis, cell proliferation, tissue regeneration, wound healing, etc. Clinically, the presence of protein markers, termed “Acute Phase Reactants (APRs)”, in the serum and plasma would be indicative of ongoing inflammation in the body. Cytokines, such as Interleukins (IL) 1 and 6 and Tumor Necrosis Factor (TNF, often referred to as TNF-α), that are produced by immune cells, stimulate the synthesis of most of the APRs in hepatocytes, which are then released into the bloodstream [1]. Some of these pro-inflammatory markers, such as TNF-α and IL-6, were also found to be elevated in chronic inflammatory conditions, not only in acute conditions.

For more than the past five decades, there has been a constant increase in the number of articles that were published on TNF-α, which would underscore its clinical significance in different disease conditions. Although several cell types in the body can produce TNF-α, in inflammatory conditions, cells from the peripheral blood such as lymphocytes, monocytes, macrophages, and neutrophils produce it in quantities that can denote an ongoing inflammation in different disease conditions [2,3]. Of note, in cardiac stress conditions, cardiomyocytes can also produce TNF-α in quantities that are sufficient in elevating cardiac inflammation [4]. Increased levels of TNF-α have been reported in several immunological diseases, and its prominent role was observed in various human diseases such as different types of arthritis, inflammatory bowel disease, bacterial and viral infections, insulin resistance and type II diabetes, Alzheimer’s, cancer, etc. Essentially, it is the pro-inflammatory role of TNF-α that is observed in many of these immune-associated diseases and disorders. Synthetic drugs and monoclonal antibodies, which can prevent TNF-α-mediated molecular mechanisms, have been approved by the FDA and are currently used in the treatment of diseases such as Crohn’s disease, Psoriatic arthritis, Rheumatoid arthritis, Juvenile idiopathic arthritis, Ankylosing spondylitis, Ulcerative colitis, Uveitis, and Hidradenitis suppurativa [5]. In addition, several off-label indications of these TNF-α-targeting drugs are also considered by the practitioners.

Some cardiovascular diseases, such as heart attack and coronary artery disease, which occur due to atherosclerosis and restenosis, are immunologically regulated. TNF-α has been reported to exacerbate both atherosclerosis and restenosis, and several studies have identified the pro-inflammatory role of TNF-α in these cardiovascular disease conditions [6,7,8]. Although there is ample evidence which demonstrates that inhibiting TNF-α can be effective in treating some cardiovascular diseases, the use of TNF inhibitors in these disease conditions is not yet approved. This review article highlights the prognostic role of TNF-α and different members of the TNF superfamily of proteins in vascular restenosis resulting from hyperplasia of vascular smooth muscle cells (VSMCs). More precisely, the role of TNF-α in regulating molecular mechanisms, specifically in VSMCs, and briefly, important details about the roles of select members of the TNF superfamily of proteins in cardiovascular complications and cellular events that are associated with disease development, are discussed. Due to the vast nature of the topic, stenosis in atheromatous plaques, in-stent restenosis, and immunomodulatory roles of all the TNF superfamily of proteins are discussed minimally when found relevant.

## 2. Cellular Events Regulated by TNF-α in Vessel Walls

As an acute phase reactant, TNF-α is amongst the early induced markers whose expression is seen almost instantaneously. In animal studies, the protein levels of TNF-α were seen elevated within hours of stimulation with endotoxins, as noticed in acute lung, kidney, or liver injury [9,10,11]. Though initially identified as a factor inducing necrosis in mouse tumors, the current understanding of TNF-α projects its multitudinous, diverse, and opposing role in normal and pathological conditions. The proactive role of TNF-α in modulating important cellular events has been reported, which drives the pathological state of the disease. Besides the below detailed cellular events that are associated with vascular restenosis, other events and processes such as leukocyte recruitment, cell adhesion, necroptosis, apoptosis, etc., also contribute to restenosis and other cardiovascular complications in which TNF-α plays a significant role in inducing pathophysiological changes. Figure 1 summarizes some of the cellular events in restenosis and the associated molecular pathways where TNF-α specifically induces or inhibits pathway intermediary proteins.

### 2.1. Cell Survival

Among the early mechanisms induced in a cell in response to stress or stimuli that affect its normal functions are the cell survival mechanisms. Cell survival mechanisms are primarily the protective mechanisms that initiate molecular pathways in the cell which help in recovering the cell from the damage that occurred. Some of the events that are known to initiate cell survival mechanisms include ER stress, unfolded protein response, oxidative damage, UV-induced DNA damage, and heat shock, etc. 

At the molecular level, TNF-α was found to promote cell survival in VSMCs while simultaneously inducing apoptosis in endothelial cells within the atherosclerotic lesions [12]. It was interesting to know that TNF-α stimulates either apoptosis or cell survival by regulating a common Rb-E2F pathway. In promoting cell survival, TNF-α favors the dissociation of E2F with the tumor suppressor protein Rb (retinoblastoma), and this results in its binding to promoters of proteins that cause active cell proliferation. Likewise, by regulating signaling mechanisms with two of its canonical receptors, TNFRI and TNFRII, TNF-α mediates cellular functions that at times contrast each other, and these are discussed in detail in later sections below. Another prominent role of TNF-α in promoting cell survival in human atherosclerotic VSMCs was through inducing cellular autophagy [13,14]. The above reports project opposing roles of TNF-α, where it promotes direct and indirect mechanisms to facilitate cell survival in VSMCs. 

### 2.2. Cell Proliferation and Migration

During restenosis, both migration and proliferation of VSMCs occur, which leads to cellular hyperplasia, and this increases the cell number at the lesion site. Recently, the role of TNF-α-mediated signaling in some aspects of heart disease was recently reviewed [15]. Shear stress remains as the most common causative factor for atherosclerosis and restenosis in arteries, which was observed both in humans and in animal models. Studies from the TNF-α-knockout mouse model were able to decipher a clear role of TNF-α in exacerbating restenosis. In TNF-α-knockout mice after carotid artery ligation, neointimal hyperplasia was found to be greatly reduced, suggesting that the use of TNF-α antagonists can help to treat restenosis in the arteries [16]. In evaluating the effectiveness of a high cholesterol diet in inducing atherosclerosis and restenosis in swine, a correlation between hyperlipidemia and the occurrence of neointimal hyperplasia was observed [17]. These in vivo observations were further supported by the in vitro studies which demonstrated reduced foam cell formation in THP-1 cells when treated with monoclonal antibodies that target TNF-α [18]. 

TNF-α seems to regulate VSMC proliferation through multiple molecular pathways, including epigenetically regulated pathways. Co-stimulatory effects of TNF-α in promoting VSMC proliferation has been previously reported [6,7,8,17,19]. In VSMCs, knockdown of Histone deacetylases, HDAC2 or HDAC10 seems to prevent cell proliferation to the same extent as in TNF-α + IGF-1 treatment groups, indicating that multiple molecular pathways may be associated with the cell proliferation in VSMCs. This also suggests that the mechanisms of TNF-α-mediated cell proliferation can also be active, independent of HDAC activity. As reported in the above reports, the other major epigenetic mediator, DNMT1, also seems to play an important role in VSMC proliferation which is greatly influenced by TNF-α.

A direct role of TNF-α in inducing reactive oxygen species (ROS) in cellular events that promote intimal hyperplasia was also evident. In vivo studies in TNF-α-knockout mice showed improved cardiac functions and decreased ROS production [20]. The oxidative damage from ROS was found to alter many cellular functions, including VSMC migration, and a clear role of TNF-α in inducing ROS in VSMCs is detailed below. Different pathways have been reported in the literature that can trigger the migration of VSMCs, and the involvement of different Kinases in promoting VSMC migration was described in detail [21]. Notably, increased production of ROS by TNF-α and the subsequent induction of matrix metalloproteinases (MMPs) was shown to promote the migration of VSMCs [22]. Since ROS promotes vascular aneurysms and restenosis, inhibition of ROS is construed as an effective strategy to prevent cardiovascular complications. In support of this, in rats, the use of the ROS quencher “Resveratrol” was found to prevent vascular injury resulting from ischemia and reperfusion by inhibiting TNF-α-induced necroptosis [23]. Furthermore, clearer evidence on the role of resveratrol was reported in rabbits where resveratrol was found to prevent restenosis in the iliac arteries by inhibiting intimal hyperplasia [24].

### 2.3. Phenotype Switch and Cell Differentiation

Understanding the functional properties and different molecular pathways initiated by proteins that are secreted from synthetic VSMCs suggests that matrix remodeling, cellular reprogramming, cytoskeletal changes, hypoxia, and the pro-angiogenic environment, promote proliferation and migration of VSMCs, are some of the important regulators of cellular processes that enhance vascular restenosis. Restenosis can also be seen as a proliferative disorder, and two of the most prominent events that occur during restenosis are cellular de-differentiation and phenotype switch. Though phenotype switch in VSMCs has direct correlations with vascular restenosis, in essence, it is also an important repair and restoration mechanism that is initiated to circumvent the damage that occurred to the vessel wall. Therefore, phenotype switch in VSMCs can be a net result of multiple molecular pathways involving other cell types from the immune system, which suggests that cellular crosstalk mechanisms also play an important role in regulating phenotype switch. With a multi-faceted role, TNF-α undoubtedly has the potential to initiate different molecular mechanisms in different cell types at the same time, which may cumulatively contribute to restenosis in the vessel walls.

During restenosis, the newly proliferated VSMCs present a prominent change in their morphology and cellular phenotype with distinctive protein expression patterns. VSMCs in healthy blood vessels can be physically or chemically stimulated to divide. Upon division, the newly formed VSMCs present a distinguishable phenotype, and this transition in the cellular phenotype can be seen both in the atheroma as well as in the restenotic lesions in vascular walls. Within the medial layers of the vessel walls, the otherwise healthy VSMCs have a characteristic contractile phenotype and normally remain in a quiescent state. On the contrary, induced by the stimulus, VSMCs that undergo a phenotype switch present a distinguishable synthetic phenotype with migratory and proliferative properties. It was previously shown that, depending on the phenotype of the VSMCs, TNF-α either induces proliferation or apoptosis in them [25]. In addition to promoting the differentiation of VSMCs into foam cells, TNF-α alone seems to contribute to vascular wall thickening and is independent of the presence or type of LDL (low-density lipoproteins) in serum. In addition to the above ex vivo culture model, 3D cell culture methods have also demonstrated disease mechanisms and provided valuable insights, although they may not reflect true in vivo conditions [26,27]. 

## 3. Molecular Mechanisms of TNF-α in Restenosis

Molecular mechanisms that are initiated by TNF-α in promoting restenosis are currently an active research area to which epigenetic regulations that stem from TNF-α-induced effects have further broadened the scope of research. In addition to the TNF-α-mediated signaling and its regulation of some of the molecular pathways that are discussed below, TNF-α-mediated mechanisms also lead to leukocyte activation, production of inflammatory cytokines, matrix remodeling, activation of transcription factors, etc., which potentially influence restenosis in the vessel walls. 

Impacting the cells with a robust stimulus to transition them from their quiescent state to a proliferative state is the basis for many proliferative disorders, including cellular hyperplasia and restenosis. In promoting restenosis, TNF-α seems to activate multiple signaling intermediates such as ERK, p38MAPK, and PI3K kinases, which are well-known for their role in promoting several pathological conditions where active cell proliferation and migration are key characteristic features [21,28]. Furthermore, a notable transcription factor that is considered a master regulator in many pathological conditions is NF-κB, which was also found to be activated by TNF-α [29]. In fact, it was shown that activation of NF-κB is also required for TNF-α to initiate migration of VSMCs, which plays a vital role in restenosis [30]. 

TNF-α is among the very few proteins that can activate their own transcription in an autocrine mode of action. Although the two specific receptors of TNF-α, TNFRII and TNFRI, are known to exert their unique and distinguishable mechanisms of action with different downstream signaling mediators, the TNFRII receptor is known to have a higher specificity towards its circulating ligand on the cell surface. In addition, TNFRII also has a subsidiary role where it transduces the signal to TNFRI, which is a unique feature of TNFRII that is associated with outside-in signal transduction. In deciphering the molecular mechanisms, it is imperative that, like many other protein markers, important information about the molecular mechanisms of TNF-α and its potential upstream or downstream targets can be inferred from in vitro studies. Earlier, a comprehensive study unraveling the clinical significance of TNF-α in promoting restenosis was conducted. The study analyzed more than three thousand patients who underwent percutaneous coronary intervention (PCI) procedures. Correlating the observations in this cohort of patients, detailed investigations were then conducted using in vivo animal studies with TNF-α-knockout mice which not only gave extensive insights on the clinical significance of TNF-α in restenosis, but also provided firm evidence that TNF-α inhibition can be a potential strategy to address restenosis [31]. Several other studies have also investigated the role of TNF-α in exacerbating different cardiovascular complications, and below are a few select molecular mechanisms that are mediated by TNF-α in VSMCs during restenosis.

### 3.1. ROS Production by TNF-α

One of the mechanisms that was shown to cause restenosis was increasing the production of ROS in VSMCs, which is mediated through NADPH oxidase 1 (Nox1), an enzyme that catalyzes the production of superoxide radicals within the cells. Smooth muscle specific knockdown of Nox1 was found to affect both proliferation and migration of VSMCs, although migration was noticed to be affected at a much higher rate [32]. Through TNF-α-TNFR1 receptor–ligand interaction within the endosomes, a drastic increase in ROS production by Nox1 was reported which was not observed in Nox1 knockout cells [33]. One of the six matricellular proteins in the CCN family, CCN1, was found to promote Nox1-mediated ROS production to induce cellular senescence [34]. In macrophages, TNF-α was reported to induce expression of CCN1 to upregulate ROS production and subsequent oxidative inactivation of JNK phosphatases to cause apoptotic cell death [35]. Interestingly, another member of the CCN family, CCN4, which is a Wnt-induced secreted protein, was reported to be induced by TNF-α, and it was found to efficiently enhance VSMC adhesion, migration, proliferation, and phenotype transition, suggesting its proactive role in promoting restenosis in the vessel walls [36]. 

An in vitro method that was found to mimic mechanical stress conditions similar to hypertension is cyclic stretch. In VSMCs, a 10% cyclic stretch at 1 Hz was reported to induce Nox1 expression which was mediated by the MEF2B protein. Nox1 expression and subsequent ROS production during cyclic stretch was found to promote phenotype switch in VSMCs. This transition from contractile to synthetic phenotype in VSMCs is accompanied by changes in cellular proteome. Notably, significant upregulation of osteopontin and simultaneously downregulation of calponin1 and smoothelin B has been reported. The same study also showed that Nox1 activation would lead to decreased density of F-actin fibers, augmented MMP-9 activity, and increased VSMC migration, which all contribute to restenosis [37]. On the same note, angioplasty, which is an interventional procedure performed to restore patency in atheromatous blood vessels also induces mechanical stretch in vessel walls which causes neointimal hyperplasia and vascular restenosis.

### 3.2. TNF-α-Induced Epigenetic Changes during Restenosis

DNA methyltransferases (DNMTs) and HDACs are two well-known classes of cellular proteins that are normally localized in the nucleus and regulate epigenetic gene expression in cells. Different isoforms of these proteins not only show differential expression patterns among various cell and tissue types, but also express predominantly in specific pathological conditions, making them unique targets for disease treatment [38]. As an early induced marker of inflammation, TNF-α, for its multifaceted role, can be anticipated to trigger the expression of several proteins, including epigenetic mediators. 

Focal adhesion kinase (FAK) is a known intracellular cell proliferating agent that aggressively promotes cell proliferation in many disease conditions, including cancer. In mice, wire injury was shown to activate FAK expression in VSMCs in the vessel wall. Interestingly, pharmacological inhibition of FAK was reported to lower the expression of DNMT3a, and this leads to hypomethylation of the DNA sequences in a few of the important gene promoters that encode contractile proteins. As a result, the expression of the contractile proteins gets elevated [39]. The study clearly showed that injury to the vessel wall would elevate FAK-mediated proliferation and the transition of VSMCs towards a synthetic phenotype, which is the underlying cause of restenosis. Importantly, this phenotype switch in VSMCs was mediated by the TNF-α receptor associated factor 6 (TRAF6). Supporting the above study, a more direct role of FAK in promoting TNF-α-mediated inflammation in the vascular endothelium of mice that underwent carotid artery ligation was reported [40]. Taken together, the above reports present firm evidence on the role of FAK in promoting vascular restenosis.

A prominent role of TNF-α in initiating epigenetic mechanisms during vascular restenosis was shown to be mediated through DNMT1, which is a robust and major DNA methyltransferase protein associated with many proliferative disorders. In the presence of IGF-1, TNF-α was reported to negatively regulate the expression of a tumor suppressor protein, SOCS3 (suppressor of cytokine signaling 3), which is mediated through promoter hypermethylation by DNMT1 [17]. The co-stimulatory effects of IGF-1, along with TNF-α, were shown to initiate different epigenetic mechanisms. Two such mechanisms were reported which detail the role of TNF-α in initiating epigenetic regulations that promote cell proliferation in VSMCs and contribute to the development of restenosis [6,7,8]. In the above referenced reports, a clear role of both the epigenetic mediators, DNMT1 and DNMT3a, in the development of restenosis was evident. One of the important questions that was addressed is: when DNMT1 is functional, would the role of DNMT3a be redundant? It was demonstrated that, when DNMT1 was silenced in VSMCs, the expression levels of the subsidiary methylase DNMT3a was increased, and this inter-regulation between the two methyltransferases could be a compensatory mechanism initiated for cell survival, to sustain normal cellular functions [8].

In addition to inducing the expression of DNMTs, TNF-α was also reported to regulate the expression of other classes of epigenetic proteins, HDACs. In the presence of IGF-1, TNF-α was shown to induce the expression of both HDAC2 and HDAC10 [19]. One of the known functions of HDAC10 is the induction of cellular autophagy. Mechanistic details on how HDAC10 promotes autophagy in VSMCs upon stimulation with TNF-α and IGF-1 are discussed in detail in the later sections. Another group of HDAC proteins are sirtuins, which are deacetylases that are grouped as a separate class of HDAC proteins, class III HDACs. One of the sirtuin isoforms, SIRT1, was found to functionally alleviate the pro-inflammatory effects of TNF-α in VSMCs [41]. Mechanistically, TNF-α prevents the expression of SM22a by promoting trimethylation at H3K27 residue in its promoter region. SM22a is needed for Casein Kinase II to activate SIRT1; thus, in the presence of TNF-α, SIRT1 expression was seen to be downregulated. Since SM22a is a marker for VSMC’s contractile phenotype, its downregulation would favor the phenotype switch to synthetic forms, which are the predominant VSMC cell types seen in the restenotic lesions. 

Another common cardiovascular complication associated with the defective functions of VSMCs are aneurysms that can occur in almost any artery in the body. Aneurysms can be seen through radiographic imaging, and they typically appear as saccular distention of the vessel walls. Notable findings associated with aneurysms include cellular senescence with a decreased number of VSMCs in the vessel wall, phenotype switch in VSMCs, vascular inflammation, matrix degradation and remodeling with MMP activity, elevated ROS, and defective autophagy [42]. As detailed above, most of these histopathological features are influenced by TNF-α. The expression of SIRT1 was often seen to be repressed in the aneurysm lesion site. This was supported by the studies that explored VSMC-specific SIRT1 knockdown, where an increased aneurysm was observed. With age and cellular senescence being major risk factors, the use of SIRT1 agonists in older patients seems to help in treating this disorder [43]. Another molecular mechanism by which TNF-α appears to be involved in enhancing aneurysms is by promoting the expression of Kruppel-like transcription factor 4 (KLF4). The binding of KLF4 and pELK1 facilitates the recruitment of HDAC2 to the G/C repressor elements of the SM22a promoter, leading to its gene silencing and phenotype switch [44]. Clearly, TNF-α seems to efficiently prevent the expression of contractile phenotype markers of VSMCs, thereby enabling phenotype transition in them and facilitating the formation of aneurysms in the cranial arteries [45].

### 3.3. TNF-α-Induced Autophagy Mechanisms

The role of TNF-α in inducing the expression of autophagy-promoting proteins, such as Beclin-1 and MAPCL-3 (also called LC3), has been reported in the past [13]. Recently, in VSMCs cultured from mouse aorta, IGF-1 treatment elevated the expression of the autophagy marker LC3-II to protect the cells from apoptosis [14]. As discussed in the above sections, it was clear that IGF-1 induces co-stimulatory effects in VSMCs along with TNF-α. One of the isoforms of HDAC proteins, HDAC10, was reported as a regulator of cellular autophagy. In the presence of IGF-1, TNF-α promotes the activity of HDAC10 in VSMCs, and this can be seen as a protective mechanism associated with cell proliferation and cell survival [19]. Here, the expression of the autophagy inducer HDAC10 appears to be dependent on the epigenetic mediator HDAC2, because the inhibition of HDAC2 was found to affect the expression of HDAC10 and not vice versa. However, it should be noted that since HDAC2 precedes HDAC10 activity early in the epigenetic pathway, HDAC2 may not have a direct role in regulating the mechanisms that induce autophagy in VSMCs.

## 4. Role of TNF Superfamily Proteins in Restenosis

The TNF superfamily of proteins consists of 19 different ligands and over 30 different receptors. These ligands have a common structural motif that helps in ligand trimerization. This trimeric subunit then interacts with the TNF-α receptor [46,47]. This ligand-receptor-mediated signaling among the TNF superfamily of proteins is one of the most studied mechanisms that drive different outside-in and intracellular signaling mechanisms. 

Contrary to the IGF-1-mediated co-stimulatory effects of TNF-α that are discussed in later sections, the molecular signaling of the Fibroblast Growth Factor inducible 14 (FN14) protein, which acts as a cognitive receptor for one of the members of the TNF superfamily of proteins, TWEAK (TNF-α weak inducer of apoptosis), was found to compromise VSMC cell survival. In this mechanism, the cleaved segment of the Plasminogen activator inhibitor-1 protein was shown to induce apoptosis in both cultured VSMCs and also in vivo in a rat carotid artery ligation model which had reduced intimal hyperplasia [48]. 

One of the other members of the TNF superfamily of proteins, tumor necrosis factor (TNF)-like protein 1A (TL1A), was found to confer protective effects on VSMCs that may help in preventing restenosis and calcification. TL1A was reported to promote the expression of α smooth muscle actin, which is a marker for contractile phenotype. At the same time, it inhibits the expression of osteopontin, which is a marker for synthetic and osteogenic phenotypes [49]. Other reports also suggest a more dependent role of TL1A in modulating VSMC functions, as was reported in patients with pre-existing conditions such as atherosclerosis, rheumatoid arthritis, aneurysms, and other vascular inflammatory diseases [50,51].

Another member of the TNF superfamily of proteins, CD137, which is also referred to as TNF receptor superfamily member 9 (TNFRSF9), was reported for its role in promoting neointimal growth [52]. In an in vitro culture system, CD137 treatment to VSMCs led to an elevated expression of NFATc1 (Nuclear Factor of Activated T lymphocytes) and vimentin, with a simultaneous increase in cell migration and proliferation. In another study, the mechanical stress from balloon injury in rat carotid arteries was found to induce NFATc1 expression which is dependent on cyclin A and the growth factor PDGF-BB [53]. Further, the ApoE null mice, when treated with CD137, presented an increased neointimal growth [52]. Phenotype switch in VSMCs was also observed with CD137 treatment to VSMCs, which showed the inhibition of specific proteins which are markers of contractile phenotype, along with elevated levels of protein markers corresponding to synthetic phenotype. These reports suggest a direct role of CD137 in promoting restenosis.

In addition to the TNF superfamily of proteins, different growth factors and inflammatory cytokines and oxidative damage in VSMCs have also been found to induce phenotype switch in VSMCs. Subsequent to transitioning into a synthetic phenotype, VSMCs were found to secrete proteins such as MMP-2, ICAM-1, PDGF, FGF-2, IL-6, IL-18, and MCP-1, which may denote them as potential markers associated with phenotype switch [54,55]. In addition to the above, other mediators such as CX3CL1, cholesterol, IL-1, IFN-g, TGF-β, and NO also seem to promote phenotype switch in VSMCs, which cumulatively indicates that TNF-α is not an individual player in regulating restenosis. On similar lines of deliberations, an aspect that is not discussed here in this article is the phenotype transition in VSMCs to osteogenic phenotype which is associated with calcification in cardiovascular diseases. Of note, calcified atheromatous plaques in advanced stages and valvular stenosis resulting from calcification are the two notable cardiovascular complications that were also influenced by TNF-α [56,57]. Several other molecular mediators that are not related to TNF-α but have direct and indirect roles in regulating phenotype switch in VSMCs have been recently reviewed [58].

## 5. Stenosis Consequent to Interventional Procedures

The revised 2005 guidelines from the American College of Cardiology (ACC), American Heart Association (AHA), and Society for Cardiovascular Angiography and Interventions (SCAI), refer to PCIs with a much broader inference [59]. As per the new guidelines, several related technologies are now considered to be components of PCIs, which include Percutaneous Transluminal Coronary Angioplasty (PTCA) and other catheter-based endovascular/intravascular interventional procedures. Most of the interventional technologies used in PCIs have the innate disadvantage of inadvertently causing damage to the blood vessels, which is not uncommon. A notable consequence often observed is neointimal hyperplasia, which results from profuse proliferation, migration, and phenotype switch in VSMCs. 

### 5.1. Vascular Stenosis Induced Due to Balloon Angioplasty

The use of balloon angioplasty remains a gold standard in restoring luminal patency in blood vessels that were occluded due to atheroma’s or other blockages. PCI technologies exist that can remove embolus efficiently. Most of these are catheter-based technologies that inevitably cause damage to the blood vessels during the procedure, such as endothelial denudation, damage to the intimal layers, and injury due to balloon distention. Vascular inflammation occurring due to these interventional procedures triggers cellular and molecular responses that can cause restenosis.

Besides inducing the expression of pro-inflammatory mediators, TNF-α also induces the expression of the nuclear receptor Nurr1, which is an early response gene whose expression was reported in in-stent restenosis. Interestingly, localized overexpression of Nurr1 in the carotid arteries of ApoE null mice prevented the formation of restenotic lesions. In cultured human VSMCs, overexpression of Nurr1 reduced the transcript levels of TNF-α, suggesting a feedback regulation involved in controlling inflammation [60]. Moreover, the CCN1 protein that was discussed in earlier sections on its role in promoting ROS production, was also found to promote VSMC proliferation and survival. Subsequent to its expression induced by TNF-α, CCN1 mechanistically binds to a6b1 integrins on the cell surface and mediates outside-in signaling. In a rat model study on restenosis occurring due to balloon injury in the carotid artery, lentiviral-mediated delivery of CCN1-specific siRNA prevented the formation of neointimal hyperplasia after balloon injury. However, the suppression of CCN1 did not had any effect on PDGF or TNF-α [61]. The same study also showed that, upon treatment with recombinant CCN1 protein to VSMCs in which CCN1 expression was knocked down, BrdU uptake was increased, which indicates an increase in VSMC proliferation.

Increase in the serum levels of the pro-inflammatory protein TREM1 (Triggering Receptor Expressed on Myeloid cells) was positively associated with the development of in-stent restenosis [11]. The same study also showed that TREM1 expression gets elevated in the neointimal and medial layers of the mouse carotid artery when it remained ligated for 3 weeks. Also, when human aortic SMCs were treated with LPS, the levels of both TREM1 and TNF-α were seen to be elevated. Interestingly, the human saphenous vein when cultured ex vivo for 7 days, thickening of the medial layer was observed, which was reduced upon treatment with a synthetic peptide LP17, which is a TREM1 inhibiting peptide. In another study, TNF-α treatment to VSMCs that were isolated from the symptomatic human carotid artery plaques led to a significant increase in the transcript levels of TREM1 which seems to be mediated through p38, JNK, and PI3K pathways. The same study also showed increased expression of the contractile marker SM22a, TREM1, MMP1, and MMP9. Notably, elevated levels of TREM1 expression was observed in the CD68+ macrophages of symptomatic plaques when compared to asymptomatic plaques, suggesting the role of MMPs and TREM1 in destabilization of the fibrous cap [62,63]. A positive feedback loop seems to be involved here in amplifying the inflammation locally at the injured site in VSMCs [64]. First, along with other pro-inflammatory cytokines, TNF-α gets secreted from VSMCs in response to injury-induced pro-inflammatory stimuli. The secreted pro-inflammatory cytokines then attract immune cells such as neutrophils, monocytes, and macrophages, that express the TREM1 receptor. The activation of TREM1 receptors on these cells subsequently elevates the expression and secretion of TNF-α, and this positive feedback loop continues until a compelling pro-inflammatory milieu is established. The pro-inflammatory cytokines and PDGF-B repress the expression of contractile proteins in VSMCs, thereby stimulating cell proliferation, migration, and phenotype switch in them, which all culminate in the development of neointimal hyperplasia and restenosis. 

### 5.2. Vein Graft Stenosis in Coronary Artery Bypass Grafts 

Restenosis is an undesirable consequence that compromises the vascular lumen. However, the mechanisms that regulate restenosis in a controlled manner can potentially help in arterialization of the grafted vein and the successful establishment of a functional conduit. In coronary artery bypass graft (CABG) surgeries, saphenous veins are often used, and for these vein grafts to remain patent and functional, their arterialization is essentially required. During the process of arterialization, luminal dilation of the grafted veins may increase up to 75% with detectable morphologic changes. Matrix reorganization and tissue remodeling will also be evident, and notably, the VSMCs in the wall of the grafted veins proliferate and increase in number, and this can be seen as a beneficial factor that contributes to graft stabilization. However, it is very essential that a tight control on VSMC proliferation is maintained in these grafted veins to help in graft maturation, which may otherwise compromise luminal patency, causing graft failure. Also, other complications from graft failure can occur which are often life threatening and require urgent attention from the surgeons. 

Several studies have shown that different molecular mechanisms occur in the vein grafts. Such studies not only increased our understanding of the hemodynamics that determine the success of graft survival, but also helped in identifying important molecular mediators which paved the way for the development of specific inhibitors. One of the potent inducers of cell proliferation is the ERK kinase which was reported to promote active cell proliferation in many cell types and is also seen expressed both in normal and pathological conditions. Normally, TNF-α induces ERK activation, but the same was not seen in VSMCs from p55 null mice, which suggests that TNF-α functions are dependent on p55. However, growth factors, such as PDGF and EGF, can activate ERK independent of TNF-α, and in fact, PDGF was identified as a more potent inducer of ERK [29,65]. The protein p55 is biologically a “soluble TNF-α type I receptor”, and its presence in the serum would prevent TNF-α from binding to its cognitive receptor TNFRI. In rabbit vein graft studies, during the process of arterialization and vein adaptation, a dramatic increase in the expression levels of TNF-α was reported in the grafted veins that were experiencing low wall shear compared to high shear. This increase in TNF-α was positively correlated with the extent of neointimal hyperplasia seen in the vein grafts. However, treatment with pegylated p55 showed no significant impact on the degree of neointimal hyperplasia between the low and high shear groups, suggesting that the early mechanisms that drive vein adaptations to the shear flow, graft arterialization, and stabilization are not influenced by TNF-α, although its levels are seen greatly elevated [66]. One of the distinctive findings that deciphered a markedly dual role of TNF-α in regulating neointimal hyperplasia in the vein grafts was the receptor that is involved. It was shown that, in the endothelial cells of the vein grafts, if TNFRI-mediated signaling is active, it would potentially promote neointimal hyperplasia, while if TNFRII-mediated signaling is active it will attenuate neointimal hyperplasia [65,67].

Clearly, the above studies do not agree with each other to a certain extent, such as on the mechanistic role of TNF-α, but they do highlight the multifaceted role of TNF-α which is still not fully understood. Such multipartite observations in animal models add more to the uncertainty and thus, extensive studies on human vein grafts could help in understanding the molecular pathways that are contributing to restenosis. On a similar note, analysis of vein and arterial grafts from human cadavers showed a higher expression of TNF-α in arterial grafts compared to vein grafts [68]. A limitation that should be noted here is the time lag between the death and tissue collection, which is long enough to induce hypoxia in tissues, which could stimulate TNF-α expression. A clear role of TNF-α in causing intima and medial thickening was reported in freshly collected human saphenous vein segments that were cultured ex vivo. Upon treatment with TNF-α, VSMCs in the cultured vein segments gets activated; this not only results in increased proliferation and migration of VSMCs, but in the presence of both TNF-α and oxidized LDL, they also transition into foam cells [69].

Another cardiovascular complication seen in patients requiring routine dialysis, such as patients with severe kidney disease, is the dysfunction of arteriovenous fistulae that were surgically created by joining an artery to a vein. Of note, arteriovenous fistulae can randomly occur in the body. Most often, venous stenosis is seen in the fistulae, and one of its main attributes is the low wall shear stress which stimulates the development of neointimal hyperplasia, and may lead to fistula failure. Several complications can occur within the dysfunctional fistula which requires endovascular PCI procedures. Molecular mechanisms that contribute to the development of neointimal hyperplasia in the fistula have been recently reviewed [70]. Similar to stenosis in the vein grafts, a prominent role of TNF-α in promoting VSMC proliferation and migration, and the eventual development of neointimal hyperplasia in vein grafts were described with sufficient details [71]. Another chronic immunological disease is Crohn’s disease, and patients with this disease actively produce perianal fistulae; this patient population can benefit from fistula closure. In a retrospective study conducted on 66 patients with perianal fistulae, a strong association between high serum levels of anti-TNF-α antibodies and fistula closure was identified, suggesting another example where inhibition of TNF-α could help in treating chronic inflammatory diseases [72,73,74].

## 6. Indications and Contraindications of Using TNF-α Inhibitors to Treat Stenosis

Same cell types that are isolated from different tissues would have different functional properties, and this was evident in many cell types such as in the endothelial cells and epithelial cells. VSMCs appear to be no different than others in presenting such functional variations. Further, cellular processes, such as proliferation and apoptosis, function differently in VSMCs isolated from media of the arteries when compared to VSMCs that were isolated from media of the veins [28,75]. Also, the signaling mechanisms of TNF-α appear to have contrasting effects on different cellular processes as discussed above. One of the interesting observations was that when VSMCs were treated with the TNF-α protease inhibitors “TAPI-0 or TAPI-1”, which prevent the release of the soluble form of TNF-α, a significant decrease in EGFR activation was noted [76]. As EGFR activation is essentially required for the migration of VSMCs, inhibition of TNF-α by TAPI-0 and TAPI-1 significantly affected VSMC migration. In the development of restenosis, VSMC migration is the foremost important cellular process involved. The multipartite role of TNF-α also seems to add another layer of mechanistic and functional differences in VSMCs in pathological conditions. Since TNF-α is seen as a master inducer of inflammation with both protective and deleterious effects, its complete inhibition can be anticipated to present detrimental effects as well. As TNF-α inhibitors prevent stem cell differentiation, tissue regeneration and healing can also be affected. With such a vast degree of variability that exists in TNF-α-mediated functions, it would be important to understand the pathophysiological aspects and molecular intercepts that crosstalk between different signaling mechanisms regulated by TNF-α. This may help in selecting a better or more suitable inhibitor for treatment that can give anticipated effects. Figure 2 details some of the molecular pathways, and interpretations on the intermediary proteins from different TNF-α related malignancies that crosstalk to initiate disease mechanisms.

The degree of variability in TNF-α-mediated effects could partly explain why the drug effects of some of the TNF-α inhibitors contradict the anticipations or present adverse events. For example, when TNF-α inhibitors were used to treat patients with chronic inflammatory diseases, such as rheumatoid arthritis and Crohn’s disease, it reflected in serious cardiovascular complications with adverse outcomes, although the incidence was very rare. On the same note, the presence of a pre-existing inflammatory condition could itself be a contraindication for the use of TNF-α inhibitors. For example, the patient population with IBD who are at a higher risk of developing serious cardiovascular complications such as congestive heart failure [77,78]. Other cardiac complications such as stroke, arrhythmia, pericarditis, and ischemic heart disease have also been reported with the use of TNF-α inhibitors [79]. Furthermore, many other medications, including antibiotics and antidepressants, have cardiotoxic effects and can potentially cause heart failure [80]. In the United States, the FDA approved the use of TNF-α inhibitors for the treatment of IBD, ankylosing spondylitis, psoriatic arthritis, and rheumatoid arthritis, with a few of these drugs predicted to have heart complications as potential side effects. Currently, more than fifty TNF-α inhibitors are being evaluated in different phase trials and many more are in the pipeline with completed preclinical studies. Importantly, the five TNF blockers, Remicade (infliximab), Enbrel (etanercept), Humira (adalimumab), Cimzia (certolizumab pegol), and Simponi (golimumab), that were approved by FDA requires enhanced safety surveillance for malignancies in pediatric and young adult patients (less than 30 years of age) when prescribed. Based on the above TNF blockers, several biosimilars were developed and are currently being tested globally, which are not discussed here. In addition to these biologics, several synthetic small molecule inhibitors were also developed which specifically target TNF-α and are being tested in different phase trials. Table 1 below briefly describes a selected list of small molecule inhibitors and their indications in different disease conditions.

Based on the interpretations that can be made from this article and from the previously published reports [81], the use of TNF-α inhibitors for the treatment of vascular restenosis remains on a positive note. Considering the potential side effects of inhibiting TNF-α, more preclinical trials and additional studies that can supersede translational research are needed, which can evaluate the benefits and side effects of TNF-α inhibitors for the treatment of cardiovascular restenosis. 

**Table 1 cancers-16-01435-t001:** List of synthetic drugs that target TNF-α and their therapeutic indications.

	Drug Name	Target(s)	Mechanism	Indications/Implications	References
1	Benpyrine racemate	TNF-α	Targets TNF-α and attenuates TNF-α-induced inflammation	Liver and lung injury	[82,83,84]
2	Clodronic acid	TNF-α, IL-1β, and IL-6	Inhibits secretion of inflammatory cytokines by macrophages	Osteoporosis, vertebral fractures, hyperparathyroidism, hypercalcemia in malignancy, multiple myeloma, pain	[85]
3	CPI-1189	TNF-α	Inhibits release of TNF-α release	Sciatica and postherpetic neuralgia, and AIDS dementia complex (ADC)	[86,87]
4	Hispidol Ataquimast	COX2, TNF-α, IL-5, IL-4, Leukotrienes and GM-CSF	Inhibits the secretion of Leukotrienes, TNF-α, and GM-CSF	Chronic obstructive bronchopneumopathies	[88,89,90]
5	Hispidol C87	TNF-α	Inhibits TNF-α mediated activity by modulating TNF-TNFR interaction by affinity binding	Liver damage	[91]
6	Hispidol CDC801	TNF-α and PDE4	A potent, orally active, and dual inhibitor	Autoimmune diseases, Crohn’s disease	[92]
7	Hispidol Hemay007	TNF-α	Inhibits TNF-α induced adhesion of monocytes to colon epithelial cells	Inflammatory Bowel Disease and rheumatoid arthritis	[93,94]
8	IA-14069	TNF-α	Directly targets TNF-α	Collagen-induced arthritis	[95,96]
9	Opinercept	TNF-α	When combined with disease-modifying anti-rheumatic drugs (DMARDs) it shows superior efficacy compared to DMARDs alone	Rheumatoid arthritis	[97]
10	OPS-2071	TNF-α	Suppresses TNF-α production from T cells	Crohn’s disease, IBD	[98,99]
11	Pomalidomide	TNF-α, IL-12, IL-16, IL-1β, MIP-1α and MCP-1	Inhibits production of TNF-α	Multiple myeloma	[100]
12	Roquinimex (Linomide)	TNF-α	Reduces the secretion of TNF-α by tumor-associated macrophages. Enhance activity of T cells, NK cells, and macrophages	Antineoplastic activity	[101]
13	Tanfanercept	TNF-α	Neutralizes TNF-α activity	Dry eye disease	[102]
14	TAPI-1	TNF-α	Prevents release of the soluble forms of TNF-α, by inhibiting TACE	Arthritis	[103]
15	UTL-5G	TNF-α	Inhibits TNF-α and other factors	Reduces cisplatin-induced hepatotoxicity, nephrotoxicity, and bone marrow toxicity	[104]

## 7. Conclusions

In this manuscript, a brief overview of the multifaceted role of TNF-α was presented. Recent information from the literature that shows a prominent role of TNF-α in inducing molecular pathways, pathway mediators, epigenetic mechanisms, autophagy, and other cellular processes regulated during vascular restenosis, is outlined. The topics discussed here are anticipated to culminate in a clear understanding of the role of TNF-α in the development of neointimal hyperplasia. Research on exploring the pathological role of TNF-α has been an active subject for more than half a century, yet exploring the molecular mechanisms regulated by TNF-α is still a subject of high interest to many researchers. The field appears to be ever-increasing with several molecular pathways that crosstalk and overlap. Due to its vast role, the field appears to be ever-lurking as well, as it encompasses a wide spectrum of immunological complications. In this review, attempts were made to provide a concise picture of the role of TNF-α in cardiovascular restenosis. By far, this is not the most comprehensive review as the topic is vast and varied. The topics discussed here are focused specifically on the role of TNF-α in restenosis with its exacerbating effects on VSMCs to provide apt information based on the recently published articles. Due to the vast nature of the topic supported by the extensive literature, selective examples that can articulate and provide a better understanding of the pathological role of TNF-α are selected and discussed here. 

## Figures and Tables

**Figure 1 cancers-16-01435-f001:**
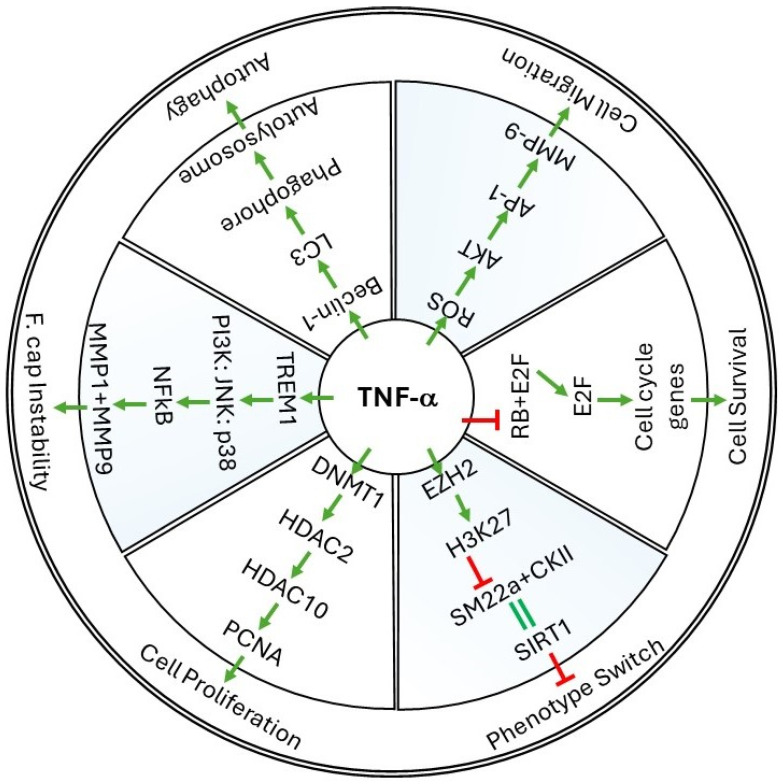
The figure is an illustration of some of the molecular mechanisms and cellular processes involved in vascular restenosis. Only a single molecular mechanism was represented for each cellular event here. Please note that multiple molecular mechanisms and crosstalks between two different pathways are involved in each of these events which are discussed in the text. F.cap represents the Fibrous cap region of the atherosclerotic plaque where the presence of thick layers of VSMCs will help protect the plaque from rupture. Green arrows indicate the favored next regulator in the pathway. Red “T” symbol indicates inhibition of the subsequent mediator. Double green lines indicate pathway mediators that promote or favor the expression of the subsequent mediator.

**Figure 2 cancers-16-01435-f002:**
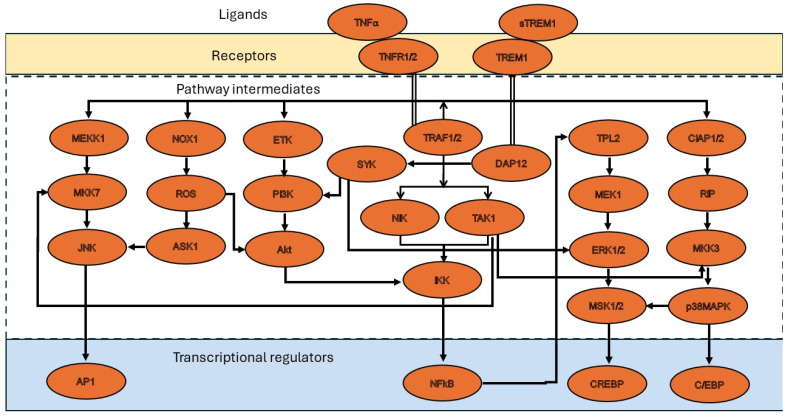
Figure shows possible crosstalk mechanisms between TNF-α regulated molecular pathways. The interpretations were made based on the reports from different pathological conditions, including cardiovascular diseases.

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
