# Peer review of "The Exacerbating Effects of the Tumor Necrosis Factor in Cardiovascular Stenosis: Intimal Hyperplasia"

_cancers, 2024, doi:10.3390/cancers16071435_

Round 1

Reviewer 1 Report

Comments and Suggestions for Authors

After reviewing this manuscript, I have determined that it is currently very incomplete and has not been carefully reviewed and verified by the authors. Since this manuscript is a review paper, it is essential to provide sufficient and useful information to readers. However, this manuscript has many typos and insufficient references, raising doubts about its reliability. Additionally, although review papers are needed to help readers fully understand the research topic, they do not present any visible graphs or tables. It is essential to present the key points of this review and basic related useful knowledge in summary tables and/or figures. Due to the incompleteness of this manuscript, I have not left all detailed comments, but only some of them are presented below.

This manuscript does not specifically describe the contents of past related studies and reports. Although this manuscript is a review and sufficient explanation and presentation of past related studies would be very important, the specific results of past related studies have not been presented (in the form of tables or figures).

Introduction: Sufficient citation of references is not being done. Are all the contents introduced in the introduction the author's own newly discovered knowledge? If it is presented based on the contents of past literature, sufficient references must be cited.

Lines 47-48: It is necessary to organize the contents of past literature on TNF-alpha in table form and describe the contents in detail.

This manuscript contains frequent English capitalization errors and typos (including in the introduction). This manuscript requires very careful and detailed review. The current state of the manuscript is considered very incomplete and difficult to be published.

Lines 58: It is recommended to present specific descriptions and drug information about synthetic drugs and monoclonal antibodies targeting TNF in a table format.

It is recommended to present the molecular mechanism correlation between TNF and vascular stenosis in the form of visual models and figures.

Line 83: “hours” A clear description is needed by presenting specific numbers. All of these parts throughout the manuscript should be replaced with explanations based on specific numbers.

The abbreviations throughout the manuscript are not organized at all.

Comments on the Quality of English Language

English very difficult to understand/incomprehensible

Author Response

Dear Reviewer 1,

Thank you for your comments and feedback. Please see the detailed response to each comment addressed and uploaded as a separate file. I appreciate your suggestions and I am hopeful that the revised manuscript has addressed the necessary concerns.

Chandra Boosani

Reviewer 2 Report

Comments and Suggestions for Authors

Chandra and Laxminarayana reviewed the prognostic role of TNF-a and different members of TNF superfamily of proteins in cardiovascular restenosis resulting from hyperplasia of smooth muscle cells. The role of TNF-a in regulating molecular mechanisms specifically in vascular smooth muscle cells (VSMCs), and important details about the roles of select members of TNF superfamily of proteins that are associated with cardiovascular complications and cellular events that are associated with disease development, are discussed. The study is potential interesting for the relevant studies, but some comments should be firstly addressed:

Major comments:

1.    It is better to contain Tables and Figures to present the detailed contents. Some contents, such as prognostic role of TNF-a and different members of TNF superfamily of proteins, should be further discussed. 

2.    The article mentioned the roles in disease development, but no relevant contents were mentioned or discussed in the text. I think the roles and potential values in complex diseases may be important for this review.

3.    It is better if TNF-a-related genes or ncRNAs could be summarized, and the interactions, especially for ncRNA-mRNA interaction, may provide more detailed reference for the relevant studies.

4.    For the TNF superfamily, other homologous members may also be critical for the role of TNF-a. Please simply discuss the other homologous genes in TNF superfamily.

5.    The whole manuscript should be carefully checked and proofed. 

Comments on the Quality of English Language

The whole manuscript should be carefully checked and proofed. 

Author Response

Dear Reviewer 2,

I appreciate the time and expert suggestions from this reviewer. I believe the revised manuscript with changes made based on all the reviewers comments, would address the concerns fully. 

Thank you for the feedback and comments.

Chandra Boosani

Reviewer 3 Report

Comments and Suggestions for Authors

In the review article "The exacerbating effects of Tumor Necrosis Factor in Cardiovascular Stenosis: Intimal Hyperplasia" the authors discuss the effects of TNF in CVS stenosis. The article is interesting to read and provides some useful insights about the topic. I have the following suggestions that I think could improve the impact of the presented article.

1. Word Choice and Clarity:

  • Ensure clarity and avoid jargon in the abstract and throughout the article for example instead of "effectuated," use "mediated," or "induced,"
  • Avoid using non-academic terms throughout the article.

2. Introduction Citations:

  • Add proper citations for all information mentioned in the introduction. This will enhance the credibility.

3. Tables and Figures:

  • Create concise tables summarizing key data points from the reviewed studies. This will improve reader comprehension and facilitate referencing specific information.
  • Include figures like graphs or diagrams summarizing the current understanding of TNF's roles and potential therapeutic approaches. Visual aids can significantly enhance the article's impact.

4. Updated References:

  • Ensure all references are up-to-date. Including research from 2023 will demonstrate the article's relevance and incorporate the latest findings.

5. Obstacles and Rationales for TNF Blockers:

  • Extend the discussion on obstacles hindering TNF blocker prescription in cardiovascular stenosis. Go beyond simply mentioning other indications or side effects.
  • Analyze specific challenges like safety concerns, lack of robust clinical data, or regulatory hurdles.
  • Provide evidence-based rationales from the literature to support the potential benefits of TNF blockers despite these challenges.
Comments on the Quality of English Language

The English writing needs to be clearer and more academic.

Author Response

Dear Reviewer 3,

We thank the reviewer for the valuable feedback and comments. In the revised manuscript, we made best efforts to address all the concerns and made changes accordingly. We hope that the changes made are agreeable to the reviewer.

Thank you,

Chandra Boosani

Round 2

Reviewer 1 Report

Comments and Suggestions for Authors

The authors have improved the manuscript more visibly than before.

However, a point that still requires improvement is to visually present the cross-talk of mechanisms and relationships between pathways in relation to the effect of TNF-a on restenosis.

The added Figure 1 is useful in that it visually confirms the mechanisms involved in TNF-a of restenosis.

However, as suggested in the caption of Figure 1, it is very unfortunate that multiple molecular mechanisms are mentioned only in text.

I highly recommend that the authors present multiple molecular mechanisms as visually as possible, as shown in Figure 1, for the reader's understanding.

Comments on the Quality of English Language

The English language status has been improved compared to before.

Author Response

  1. The authors have improved the manuscript more visibly than before.

Response: I thank the reviewer for the suggestions in improving this manuscript, and am glad to make changes as suggested.

  1. However, a point that still requires improvement is to visually present the cross-talk of mechanisms and relationships between pathways in relation to the effect of TNF-a on restenosis.

Response: As suggested by the reviewer, a new figure “Figure 2” is now included in the revised manuscript. Efforts were made to include information from the published reports on restenosis that crosstalk with other pathways which TNF-a regulates. Care was taken to make sure that presumptive or unclear information was not projected in this new figure.

  1. The added Figure 1 is useful in that it visually confirms the mechanisms involved in TNF-a of restenosis. However, as suggested in the caption of Figure 1, it is very unfortunate that multiple molecular mechanisms are mentioned only in text.

Response: Additional molecular mechanisms that were involved in restenosis are now part of the new Figure 2, included in the revised manuscript, which also details the crosstalk between the molecular pathways. Please note that to avoid any possible misrepresentation on the localization of the proteins such as NOX1 and NFkB etc. which can shuttle within the cell, cell compartments were not detailed in Figure 2 instead, the nature of the proteins was described such as ligands and receptors.

  1. I highly recommend that the authors present multiple molecular mechanisms as visually as possible, as shown in Figure 1, for the reader's understanding.

Response: I appreciate the reviewer on the insights regarding the Figure 1. Also, as identified by the reviewer in the earlier comment, sufficient details were provided in the text regarding the pathways. The readers can certainly refer to the cited articles that have full details, including visual graphics on the pathways. In addition, databases such as the Kegg database provide vast information on different TNF-a regulated pathways based on published evidence, which the reviewers and researchers are aware of. Due to the vastness and the number of pathways involved, I apologize for not being able to expand the Figure 1, beyond its current projection. I am hopeful that the reviewer understands these limitations.

Reviewer 3 Report

Comments and Suggestions for Authors

Most of my concerns were addressed appropriately.

However, I would recommend adding citation(s) of the reference for the data summarized in the Table to allow easy access to the detailed information.

Comments on the Quality of English Language

Minor revsions

Author Response

As suggested by the Reviewer 3, references were included in the table. I thank the reviewer for the positive feedback and comments that made this manuscript more presentable.

Round 3

Reviewer 1 Report

Comments and Suggestions for Authors

No more comments.